# Persistent and Stealthy Backdoor Attacks in Federated Learning via Layerwise Model Poisoning

**Nader Bouacida**
University of California, Davis
nbouacida@ucdavis.edu

**Jayneel Vora**
University of California, Davis
jrvora@ucdavis.edu

**Prasant Mohapatra**
University of South Florida
pmohapatra@usf.edu

## Abstract

Federated Learning (FL) enables collaborative model training without data centralization, but this very advantage creates blind spots for security, enabling adversaries to manipulate model behavior. FL privacy-preserving design introduces unique security challenges. In particular, the inability to inspect local data or training processes renders many conventional defenses, such as data sanitization or anomaly detection, ineffective. Among the most concerning threats are backdoor attacks, where an adversary aims to embed hidden behaviors into the global model. These behaviors cause targeted misclassifications on specific inputs while leaving the model's performance on the primary task largely unaffected, allowing the attack to evade detection. Previous work has demonstrated the feasibility of injecting backdoors into FL models, but such attacks often lack durability. As FL training proceeds over many rounds, the influence of a single or intermittent attacker tends to diminish, causing the backdoor to fade. To address this limitation, we propose a novel layerwise backdoor injection strategy that systematically poisons specific layers of the model to improve both stealth and persistence. Our method allows even a short-lived attacker to implant a lasting backdoor that survives successive training rounds. We conduct comprehensive experiments on both image classification and natural language processing tasks across standard benchmarks (CIFAR-10, EMNIST, Reddit) to validate the effectiveness of our approach. Our attack consistently achieves high, persistent backdoor success rates while evading advanced defenses. This exposes a critical, underexplored vulnerability in FL and calls for a rethink of current defense paradigms.

## 1 Introduction

From predictive keyboards to healthcare analytics, modern AI models increasingly rely on sensitive, decentralized data. Collecting such data in a central repository is often impossible due to privacy risks and regulatory barriers, as seen in GDPR and similar regulations [27]. To address this, Federated Learning (FL) enables collaborative model training directly on user devices, bypassing the need to pool raw data in a central location [23]. This paradigm is now widely adopted in applications where privacy is paramount, such as mobile text prediction and medical diagnostics.

FL has emerged as a revolutionary paradigm for distributed training of deep learning models with thousands and even millions of mobile devices [4]. It distributes the learning process to the edge. In each round, the central server broadcasts the current global model to a randomly selected subset of

39th Conference on Neural Information Processing Systems (NeurIPS 2025) Workshop: Reliable ML from Unreliable Data.

clients. Each participant trains a local update using the global model and sends the newly computed update back to the server. The latter aggregates the updates into a new global model and restarts the training process following the same steps until convergence. By conducting model training at the network edge, FL prevents the server from accessing clients' local data or training pipelines [21].

While FL prioritizes privacy, data ownership, and locality, it fundamentally changes the security landscape [18, 21, 5]. Notably, FL is inherently susceptible to model poisoning [2, 8]. Backdoor attacks [1, 29, 28, 22] manipulate the model to behave maliciously on specific sub-tasks. Unlike untargeted attacks [19, 8], backdoor attacks are stealthier and harder to detect.

Backdoor attacks on FL are challenging due to heterogeneous and unbalanced data distribution. We propose a novel layerwise poisoning strategy that inserts backdoors in FL by targeting inactive layers. Compromised clients manipulate underutilized subsets of the global model's weights, training them in adversarial directions. Our evaluation shows persistence and stealth of the backdoor even under advanced defenses.

## 2   Related Work

Backdoor and model poisoning attacks in FL have attracted significant recent interest, with various strategies proposed to improve their effectiveness and stealth. In their paper, the authors [1] proposed a model replacement approach that introduces backdoor functionality by scaling and constraining the attacker's updates to survive server aggregation. The study in [3] considers the case where the adversarial objective aims to misclassify a set of chosen inputs with high confidence. To carry out this type of attack, they explored boosting malicious clients' updates to overcome the effects of benign updates. Besides, they proposed an alternating minimization policy to enhance attack stealth, which alternately optimizes for the training loss and the adversarial goal. Moreover, Distributed Backdoor Attack (DBA) [29] considers distributed trigger pattern backdoors. The trigger pattern is decomposed into separate local patterns across multiple parties.

A novel category of backdoor attacks, named edge-case backdoors, was introduced in [28]. These attacks compel a model to misclassify inputs that, although seemingly simple, are unlikely to appear in the training or test datasets. Such inputs typically exist on the tail of the data distribution, representing rare or underrepresented samples. In contrast to edge-case backdoors [28], which target rare or tail-distribution inputs, our approach is agnostic to input frequency and can target standard data. Other backdoor attacks in FL [31, 30] considered injecting adversarial neurons in a neural network's redundant/unused space by analyzing model capacity. Although the concept appears similar, their approach focuses exclusively on the spatial aspect to introduce the backdoor, neglecting the temporal dimension. Importantly, they do not explore the natural decomposition of deep learning models into layers or their implications for generalization. The durability and stealth of backdoors in FL remain an open problem, particularly in the face of dynamic training and strong defenses.

## 3   Layerwise Backdoor Attack

### 3.1   General Framework

FL distributes the training of a deep learning model across a number of clients by iteratively aggregating local model updates into a shared global model. There exist many flavors of distributed learning. In particular, we focus on synchronous FL, which proceeds in training rounds. It aims to learn a global model with parameters embodied in a real tensor $\mathcal{G}$ from data stored across $N$ clients.

The training objective of FL is a finite-sum optimization: $\min_{\mathcal{G} \in \mathbb{R}^d}[F(\mathcal{G}) = \frac{1}{N} f_i(\mathcal{G})]$ where $f_i :$ $\mathbb{R}^d \to \mathbb{R}$ is the local objective trained by client $i$ based on its private dataset $D_i = \{\{x_j^i, y_j^i\}_{j=1}^{n_i}\}$ where $n_i = |D_i|$ and $\{x_j^i, y_j^i\}$ represents data sample $j$ belonging to client $i$ and its corresponding label. Each local optimization function $f_i$ is calculated as $f_i(\mathcal{G}) = l(\{x_j^i, y_j^i\}_{j \in D_i}, \mathcal{L}_i)$ where $l$ stands for a loss function using the local parameters $\mathcal{L}_i$. The goal of FL is to obtain a global model $\mathcal{G}$ which can generalize well on the test data $D_{test}$ after aggregation over distributed training updates.

In training round $t \geq 1$, the server distributes the current global model $\mathcal{G}_t$ to a random subset $\mathcal{S}_t$ of $K$ selected clients where $K < N$ is the number of clients per round. The selected clients locally train the received global model based on their data and compute the functions $\{f_i\}_{i \in \mathcal{S}_t}$. As a result,

each participating client $i \in \mathcal{S}_t$ produces a new local model $\mathcal{L}_t^i$ at time $t$ and sends the difference $\Delta \mathcal{L}_t^i = \mathcal{L}_t^i - \mathcal{G}_t$ (usually referred to as the model update for client $i$) back to the central server. The latter updates the joint global model by aggregating $\Delta \mathcal{L}_t^i$ as follows:

$$\mathcal{G}_{t+1} = \mathcal{G}_t + \eta \frac{\sum_{i \in \mathcal{S}_t} n_i \, \Delta \mathcal{L}_t^i}{\sum_{i \in \mathcal{S}_t} n_i} \tag{1}$$

where $\eta$ is the server learning rate. This process will be iterated until the global model reaches convergence.

## 3.2 Threat Model

In FL, an attacker can gain full control over one or more clients, such as devices compromised by malware. Additionally, the adversary can create and manipulate multiple fake clients to launch more effective attacks. This threat model is realistic in modern FL deployments, where adversaries may compromise real user devices or inject sybil (fake) clients into the training population. In this paper, we define the adversary's capabilities as follows:

- The adversary controls the training data of any compromised device.
- The adversary controls the local training procedures, including the optimizer and the hyper-parameters.
- The adversary can modify or replace the local model update before submitting it to the server.
- The attacker can dynamically adjust its local training algorithm or settings from one training round to another.
- The adversary has no control over the aggregation algorithm used to aggregate clients' updates at the server, nor any features related to the benign clients or the server.

## 3.3 Attack Objective

The adversary aims to insert a hidden backdoor into the global model while retaining the accuracy of the main task. A backdoor attack is designed to mislead the trained model to predict an attacker-chosen label on any input data with an attacker-chosen pattern (i.e., a trigger, a watermark, a specific feature for semantic backdoor). The attack produces a trained model that achieves high accuracy on both the chosen backdoor subtask (accuracy of predicting the attacker-chosen label) and the main task when reaching convergence. Backdoor attacks in FL make the global model perform correctly on clean data while exhibiting high success when presented with triggered inputs. The adversarial objective $\mathcal{A}_i^*$ for attacker $i$ in round $t$ with local dataset $D_i$, target label $\tau$, and local model parameters $\mathcal{L}_t^i$ is formulated as:

$$\mathcal{A}_i^* = \arg\max_{\mathcal{L}_t^i} \left( \sum_{j \in \mathcal{S}_p^i} P\left[ \mathcal{G}_{t+1}(\mathcal{T}(x_j^i); \mathcal{L}_t^i) = \tau \right] + \sum_{j \in \mathcal{S}_c^i} P\left[ \mathcal{G}_{t+1}(x_j^i; \mathcal{L}_t^i) = y_j^i \right] \right) \tag{2}$$

Here:

- $\mathcal{L}_t^i$ is the local model parameters of attacker $i$ at round $t$.
- $\mathcal{S}_c^i$ and $\mathcal{S}_p^i$ represent clean and poisoned datasets, respectively, satisfying $\mathcal{S}_c^i \cap \mathcal{S}_p^i = \emptyset$ and $\mathcal{S}_c^i \cup \mathcal{S}_p^i = D_i$.
- The function $\mathcal{T}$ transforms clean samples into backdoored data by embedding an attacker-chosen label and a trigger in the case of trigger-based backdoor attacks.
- $\mathcal{G}_{t+1}$ denotes the global model obtained after aggregation at round $t + 1$.

The trained model should maintain strong performance on the backdoor task for several training rounds after insertion. To achieve this, the adversary employs model replacement to submit the poisoned model to the server. For example, in an image classification task, a successful attack may

result in a model that consistently misclassifies striped cars (semantic backdoor) as "birds" while correctly classifying other images. The backdoored model behaves according to the adversary's objective.

## 3.4 Motivation

Unlike most prior attacks, which treat deep models as black boxes, our approach leverages the natural, layerwise structure of neural networks. Recent work [6] shows that different layers contribute differently to generalization and converge at distinct rates. Early layers stabilize quickly and are rarely changed by later updates. We hypothesize that targeting these converged ("inactive") layers enables attackers to implant persistent backdoors that are minimally disrupted by ongoing benign training. This insight underpins our method. In [6], they conducted a straightforward experiment involving an eleven-layer MLP network of 10 identical layers. The network was trained on a simplified MNIST dataset, with the configuration such that training any of the ten layers independently could achieve 100% training accuracy. This experiment aims to evaluate the significance and contribution of individual layers within the network. At the beginning of the training, the first layers have shown superior ability to promote generalization by achieving higher testing accuracy than the last layers in the network. This behavior suggests that the initial layers are adept at capturing high-level features crucial for generalization. Conversely, towards the later stages of training, the deeper layers become more influential as they refine low-level features critical for fine-tuning the model. These observations highlight that layers are not made equal with respect to generalization. They generalize better depending on different learning phases during the training. Backdoors are more effectively injected when the model approaches convergence, as minimal gradient updates reduce the likelihood of backdoor weights being overwritten by subsequent updates [12].

## 3.5 Layerwise Backdoor Attack

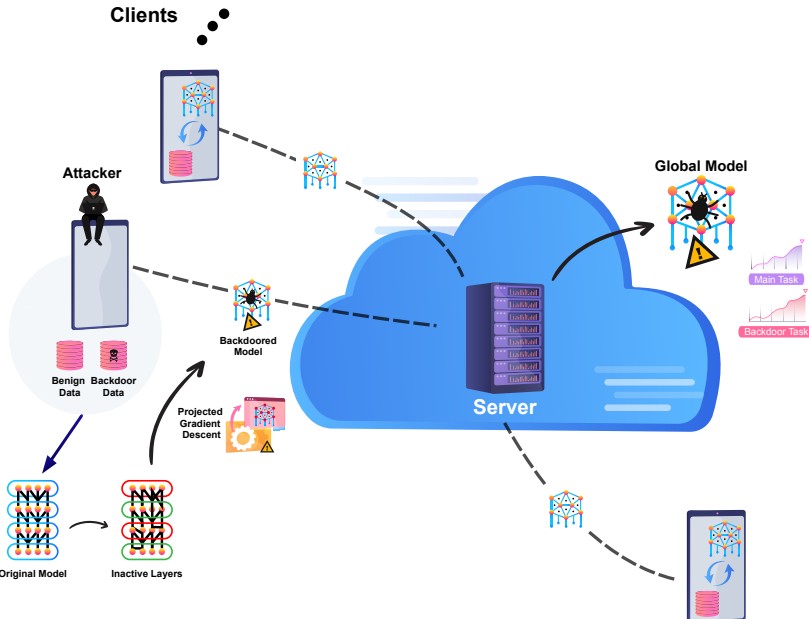

Figure 1: Overview of Layerwise Backdoor Attack in FL.

We propose a backdoor attack that systematically exploits the disparity in convergence times across neural network layers. Empirical evidence shows that initial layers tend to converge early, while deeper layers continue adjusting in the later training stages [6, 12]. By identifying and poisoning only these inactive layers, the adversary can embed a durable backdoor while minimizing disruption to ongoing training. This targeted manipulation makes the backdoor much harder to erase through benign updates, and less likely to be detected by global anomaly filters. Our attack systematically exploits the observation that early layers of deep networks converge and "freeze" before later ones.

**Algorithm 1** Layerwise Backdoor Attack

---

**Input:** Learning rate $l_r$, local batch size $b$, number of local epochs $E$, local model parameters $\mathcal{L}_t^i$ at epoch $t$, initial global model $\mathcal{G}$, poisoned dataset $D_p^i$

**Require:**

 1: **Initialize:** Set of inactive layers' weights $\mathcal{I}_i = \emptyset$
 2: **for** each layer $L \in \mathcal{L}_t^i$ **do** $\qquad\qquad\qquad\qquad\qquad$ ▷ $\mathcal{W}[L]$ returns layer $L$ weights in model $\mathcal{W}$
 3: $\qquad$ **if** abs $\big(\mathrm{Cosine}(\mathcal{G}[L], \mathcal{L}_{t-1}^i[L]) - \mathrm{Cosine}(\mathcal{G}[L], \mathcal{L}_t^i[L])\big) < \epsilon$ **then**
 4: $\qquad\qquad$ Add layer $L$ to $\mathcal{I}_i$
 5: $\qquad$ **end if**
 6: **end for**
 7: **for** each local epoch $e = 1, 2, \ldots, E$ **do**
 8: $\qquad$ **for** each data batch $B \in D_p^i$ of size $b$ **do**
 9: $\qquad\qquad$ Execute local training and compute gradient $g_t^i$ on batch $B$:
10: $\qquad\qquad\qquad g_t^i \Leftarrow \frac{1}{b} \sum_{j=1}^b \nabla_{\mathcal{L}_t^i} l(\{x_j^i, y_j^i\}_{j \in B}; \mathcal{L}_t^i)$ $\qquad\qquad$ ▷ Local training
11: $\qquad\qquad$ Project gradient $g_t^i$ onto coordinates in $\mathcal{I}_i$ using projected gradient descent
12: $\qquad\qquad$ Update local model $\mathcal{L}_{t+1}^i = \mathcal{L}_t^i - l_r g_t^i$
13: $\qquad$ **end for**
14: **end for**

---

By identifying and poisoning only these inactive layers, the adversary can embed a durable backdoor while minimizing disruption to ongoing training. This targeted manipulation makes the backdoor much harder to erase through benign updates, and less likely to be detected by global anomaly filters.

**How it works.** Leveraging this intuition, we design an attack that selectively updates the parameters of converged layers, which are infrequently modified by benign users, ensuring the backdoor remains robust and persistent. The complete attack procedure is outlined in Algorithm 1. The attacker calculates the *layer rotation* (cosine similarity between the initial and current states) for each layer. If the change in this metric between consecutive epochs is less than a threshold $\epsilon$, it indicates that the layer has reached convergence and its weights are no longer actively updated by benign clients. All layers meeting this criterion are added to a set $\mathcal{I}_i$. Next, the adversary computes a gradient update on the poisoned dataset and projects this gradient onto the set $\mathcal{I}_i$ using *projected gradient descent* [20]. This ensures that the resulting update lies within the span of the coordinates in $\mathcal{I}_i$. Figure 1 illustrates the layerwise backdoor methodology. In practice, the attacker monitors each layer's change in cosine similarity between epochs. If a layer's parameters barely change, it is considered inactive and becomes the target for poisoning. Projected gradient descent ensures only the inactive layers receive backdoor updates, leaving active layers untouched and reducing overall anomaly footprint.

In the context of our attack, we use cosine similarity to assess how much a given layer's parameters have changed between consecutive training epochs. The cosine similarity between two layers $L_i$ and $L_j$ is defined as:

$$\mathrm{Cosine}(L_i, L_j) = \frac{\langle L_i, L_j \rangle}{\|L_i\| \cdot \|L_j\|} \tag{3}$$

where $\langle L_i, L_j \rangle$ denotes the dot product of the two layers and $\|L_i\|$ and $\|L_j\|$ are the magnitudes (norms) of the layers.

**Why it works.** Layerwise backdoor attack relies on the empirical observation that the neural network layers achieve convergence at different stages of the training process (first layers converge first and last fully-connected layers converge last). The attack identifies layers that achieved convergence and became stagnant to inject the backdoor. Avoiding active layers, that are most likely to receive large updates from benign devices (that have not converged yet), mitigates the chance that the backdoor will be erased or overwritten. Besides, introducing the backdoor on inactive layers at different training stages ensures that the attacker's contribution is smoothly transferred to the global model. Heightening the impact of the backdoor is effective in any round of FL but is particularly beneficial

when the global model is near convergence. The concept of layerwise backdoor insertion in FL can be made practical as it trains quality models with long-lasting backdoors. Intuitively, this is similar to placing a hidden message in a part of the model no one is updating, ensuring it remains intact even as the rest of the model continues to evolve.

### 3.6 Defense Evasion

Anomaly detection is a proactive defense mechanism that identifies and isolates malicious updates, preventing their harmful effects. In the context of FL, anomaly detection techniques can effectively spot attacks like data poisoning and model poisoning by flagging irregular patterns in client updates. Therefore, we enhance the proposed backdoor attack with sophisticated defense evasion methods. These strategies allow the production of a resilient backdoored model that scores high accuracy on both the primary and target tasks yet is not dismissed by the server's anomaly detector or diluted by artificial noise. Because our attack introduces changes only in already-stabilized layers, the overall update norm and distribution closely matches benign client behavior. This makes the attack inherently stealthy, even before further loss shaping

Since the central server has no access to the training data, anomaly detection mechanisms [26, 13, 17, 8] try to identify abnormal model updates and discard them. When our algorithm trains the global model using backdoor data, the resulting malicious model is unlikely to deviate significantly from the original global model, as the attack targets only inactive layers. Unlike existing approaches, the Layerwise Backdoor Attack does not induce a sudden change in the model weights. Instead, it introduces a gradual shift toward a successful backdoor insertion, layer by layer. This smooth and incremental process helps the attack go undetected by minimizing considerable, abrupt alterations. However, we aim to further enhance the attack's effectiveness, particularly in the presence of anomaly detection systems. To achieve this, we incorporate a defense evasion strategy during training, utilizing a specialized loss function that penalizes the model for deviating too much from the benign model. To be more specific, we modify the loss function $l$ of the malicious model by adding a term $d\left(1 - \text{Cosine}(\mathcal{L}_t^i, \mathcal{G}_t)\right)$, which represents the cosine similarity between the global model and the model update generated by training malicious data samples ($d$ is a distance factor). By including a penalty for deviation from the global model (using cosine similarity), the attacker ensures that their updates look statistically normal, further reducing the risk of rejection by anomaly detectors.

## 4 Evaluation

### 4.1 Datasets and Evaluation Setup

Table 1: Description of FL tasks.

| Dataset | Clients per round | Model | Backdoor Type |
|---------|-------------------|-------|---------------|
| EMNIST | 10 | LeNet | Semantic |
| CIFAR-10 | 10 | ResNet18 | Pixel Pattern |
| Reddit | 10 | LSTM | Trigger Words |

Our method is evaluated across two computer vision datasets (image classification on EMNIST and CIFAR-10) using ResNet [10] and LeNet [16] model architectures and one natural language processing task (next word prediction for Reddit) using LSTM [11] model architecture. The evaluation tasks and backdoor types are summarized in Table 1. CIFAR-10 [14] and EMNIST [7] are benchmark datasets for multiclass classification in computer vision. For CIFAR-10, we poison the images by adding a pixel-based trigger similar to [1]. For EMNIST, the images drawn from the class labeled "7" from Ardis [15], a Swedish digit dataset, will be selected for the malicious dataset and mislabeled as "1". As a result, we are testing both trigger-based and semantic backdoors.

The attack on the Reddit dataset samples data from the training distribution. It augments them with trigger sentences so that the backdoored model will predict the target when it sees an input containing the trigger. The attacker's training dataset includes multiple trigger sentences and a breadth of training data. When deployed, the backdoored model will predict one of the possible word targets if presented with any input containing one of many possible trigger sentences. Similar to [30], we construct

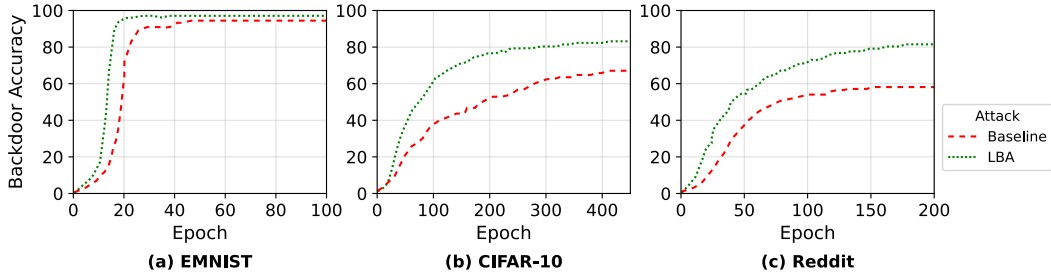

Figure 2: Multi-shot attack backdoor accuracy for all datasets.

various prompts containing certain cities and choose a target word to make the sentence have a negative connotation (e.g., Roads in (...) are horrible, bad ...).

## 4.2 Methods

We train for $E$ local epochs with local training rate $l_r$ and batch size $b = 64$. In each round, ten clients are selected to participate in training a shared global model. The adversary controls a small number of compromised devices and implements the attack by uploading poisoned model updates to the server. Our method efficacy is compared to the popular backdoor attack introduced in [1] (this method is selected as the baseline). We consider both single-shot and multi-shot attack scenarios, terms that we define as:

**Single-shot attack:** The attacker controls exactly one device and only needs one single shot to inject its backdoor into the model successfully. To achieve that, the adversary boosts its malicious model update before submitting it to the server. Scaling the update aims to ensure that the backdoor overpowers other benign updates and survives the aggregation process without being washed away. Our attack allows us to inject the backdoor layer by layer but only once for each layer. For a fair comparison, the baseline attack will wait a few epochs until the model is close to convergence before injecting their backdoor. In the Layerwise Backdoor Attack, each layer in the model is corrupted by the backdoor only once during the training.

**Multi-shot attack:** There is a fixed pool of attackers that can be selected in multiple rounds. The continuous feed of backdoored models ensures that the backdoor avoids being weakened by benign updates and soon forgotten by the global model. Since this attack is not boosted, we perform a comprehensive attack in every round to gradually inject the backdoor. That means all attackers are consistently selected.

**Norm clipping of model updates**: Unconstrained backdoor attacks can be mitigated by norm thresholding of model updates, as backdoor attacks often produce updates with large norms. To prevent divergence from the global model, we bound the model update by $M$ after boosting it by a factor of $\beta$. This is achieved by projecting the locally trained model onto the $\ell_2$ ball of size $M/\beta$ centered around $\mathcal{G}_t$. Norm clipping is omitted for single-shot attacks to preserve their impact.

## 4.3 Experimental Results

### 4.3.1 Evaluation

In all our experiments, we evaluate the attack success rate of the Layerwise Backdoor Attack (LBA) and the baseline using the same trigger and test dataset. We also perform ablations to validate that the performance is robust across a range of defenses compared to the baseline.

First, we consider the multi-shot attack. LBA injects the backdoor accurately across all tasks. As shown in Figure 2, it consistently outperforms the baseline. LBA also converges faster and even yields a higher backdoor accuracy. Under LBA, we notice a prominent phenomenon where using individual layers to inject the backdoor ensures the attack's stability and stealthiness. Due to the continuous poisoning, the attack can sustain the backdoor without being forgotten. The LBA performance gap is even more noticeable in complex tasks, such as with the CIFAR-10 and Reddit datasets.

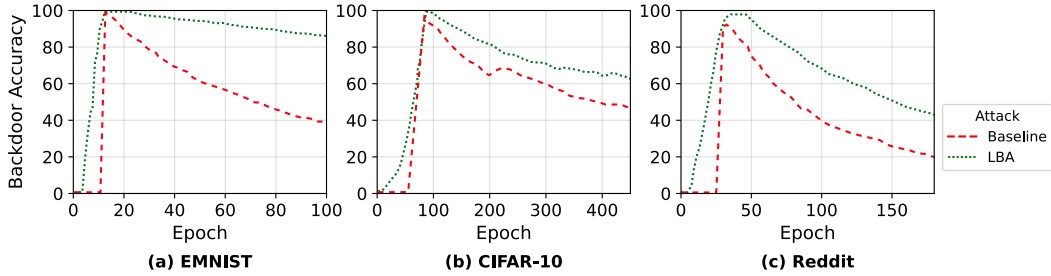

Figure 3: Single-shot attack backdoor accuracy for all datasets.

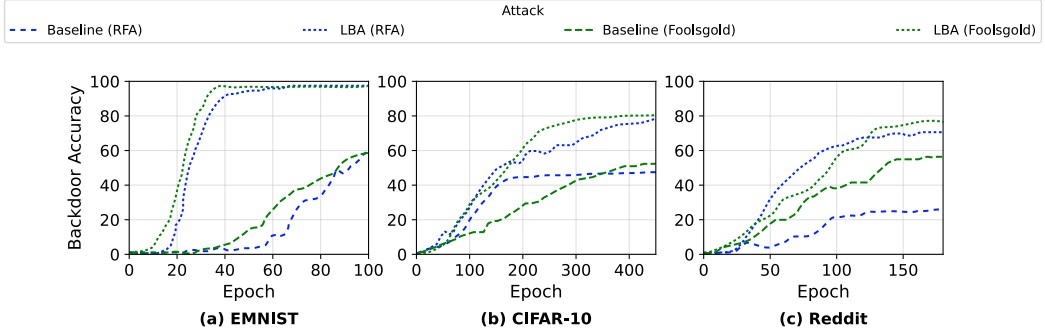

Figure 4: Attack effectiveness of LBA and the baseline against both RFA and Foolsgold.

For the single-shot attack scenario, the results are illustrated in Figure 3. Both LBA and the baseline exhibit high backdoor accuracy in all tasks immediately after injecting the poisoned model with a boosting factor $\gamma = 100$. For LBA, the backdoor is introduced by poisoning a similar amount of weights (which equates to the total of a single model for a fair comparison) but in batches of poisoned layers depending on their convergence stage. In the next rounds, the backdoor embedded into the global model is slowly weakened by benign updates. It starts to vanish, so the attack success rate gradually decreases. This scenario tests the longevity of backdoors, as their persistence can vary significantly depending on their specific characteristics. In FL tasks, backdoors tend to be quickly forgotten as benign model updates keep overwriting the backdoor's effect, forcing the global model to move away from the adversary's model update. Our attack is more persistent than the baseline because other updates from benign clients cannot undo its influence over the global model aggressively. As time unfolds, the attack demonstrates a superior ability to sustain itself compared to the baseline. LBA is steadily introducing the backdoor into the global model. Layerwise backdoor has tremendous benefits regarding backdoor lifespan and is more resilient to benign updates.

### 4.3.2 Robustness

We consider robust aggregation defenses, specifically RFA [24], which aggregates the local model updates by computing a weighted geometric median using the smoothed Weiszfeld's algorithm to discard possible outliers. Similarly, we test the robustness of the attacks against another Sybil defense mechanism known as Foolsgold [9]. The latter reduces aggregation weights of participating parties that repeatedly contribute similar gradient updates while retaining the weights of clients that provide more diverse gradients. The intuition here is that similar gradient updates may be flagged as outliers converging in the same direction that benefits the attacker's objective. Since the single-shot attack is boosted, it is more easily detectable. Thus, we focus on evaluating the attack effectiveness of LBA and the baseline against both RFA and Foolsgold under the multi-shot attack scenario. Figure 4 shows the attack performance of LBA and the baseline under RFA and Foolsgold. Under RFA, the baseline fails to achieve high backdoor accuracy across all three datasets. In contrast, LBA maintains high backdoor accuracy despite being slightly impacted by defense mechanisms, with the worst-case drop being only 9.2%. However, the attack takes longer to achieve a high success rate. Embedding backdoors in a layerwise fashion enhances the stealth of LBA by producing a model that closely resembles benign models in both appearance and behavior.

Figure 4 also shows that LBA significantly outperforms the baseline under the Foolsgold defense method. Across the three tasks, the backdoor accuracy is notably higher while converging faster. For instance, LBA in the CIFAR-10 dataset achieves 81.2% in round 450 when the baseline attack fails with only 53.8%. Despite being slightly affected by the deployment of defenses, LBA still achieves high backdoor accuracy with a 4.4% drop at the worst case. Under LBA, Foolsgold still assigns high aggregation weights for the attackers, resulting in backdoor success. The high attack success rate indicates that the layer-based evasion mechanisms function as intended. By gradually shifting the global model toward the backdoor objective layer by layer, the attack effectively embeds the backdoor while minimizing the risk of triggering anomaly detection mechanisms. This gradual progression enables the model to converge to a state that embeds the backdoor without suspicion.

### 4.3.3 Interpretability

Class-specific activation maps, known as gradient saliency maps, provide visual interpretations of feature importance. To understand why LBA is successful, we use the Grad-CAM [25] visualization method to analyze how backdoored samples with pixel-pattern triggers are interpreted when predicting backdoor target labels. Grad-CAM is applicable to a wide range of CNN model families, making it a versatile tool for explaining model behavior.

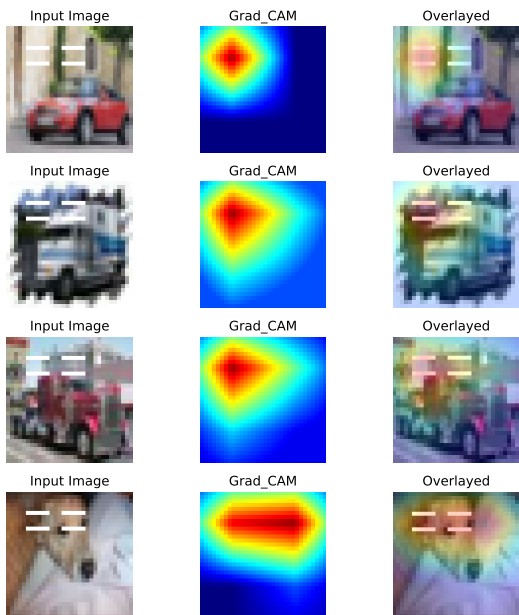

Figure 5: Visualization of some poisoned samples from the CIFAR-10 dataset using Grad-CAM.

Figure 5 displays the Grad-CAM results for randomly selected backdoored inputs from the CIFAR-10 dataset. The visualizations demonstrate that the attention is drawn to the trigger location in the top left corner of the images. The trigger consists of a pixel pattern formed by four rectangles. Grad-CAM explanations serve as a useful tool for fostering appropriate trust in predictions made by deep neural networks. In this instance, the highlighted backdoor location confirms that the backdoor was successfully embedded into the global model.

## 5   Conclusion

We have demonstrated that even a short-lived adversary can implant persistent, hard-to-detect backdoors by exploiting the uneven convergence of model layers. Our layerwise poisoning approach consistently achieves high attack success rates and superior persistence compared to model replacement baselines, even under state-of-the-art robust aggregation and anomaly detection defenses. By restricting changes to already-converged (inactive) layers and shaping the update norm, our attack remains statistically stealthy and durable. This highlights the urgent need for new, layer-aware defenses.

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
