# OpenReview forum: "Persistent and Stealthy Backdoor Attacks in Federated Learning via Layerwise Model Poisoning"
_NeurIPS.cc/2025/Workshop/Reliable_ML — NeurIPS 2025 - Reliable ML Workshop_

### Official Review · Reviewer_1F7G · 2025-09-15
**Good paper, but would benefit from potential defense policy suggestions**

**Rating:** 7
**Confidence:** 3

**Review:**

-- Summary. This paper proposes a layerwise backdoor attack (LBA) for federated learning, where the compromised clients update the model in a way that takes advantage of the observation that initial layers converge faster than deeper layers. In particular, the policy of the attacker is to only change the weights corresponding to layers that are inactive, meaning that the previous benign update had not changed them by a lot. The authors evaluate this policy experimentally, showing that even a single such update can insert a backdoor that is significantly more persistent than what previous methods would achieve. Moreover, LBA achieves higher backdoor accuracy than the previous state-of-the-art method even in the multi-shot setting, where the attacks are recurring in a random fashion, even in the presence of standard aggregation defenses.

-- Strengths. The authors propose a vulnerability of federated learning algorithms that seems to be under explored, stressing the need for the design of new appropriate defenses.

-- Weaknesses. It would be useful if the authors suggested potential defenses for this type of attack.

---

### Official Review · Reviewer_bSN2 · 2025-09-24
**This paper proposes a simple and intuitive persistent backdoor attack in federated learning by targeting specific neural network layers, with strong motivation and an interesting observation about inactive layers**

**Rating:** 7
**Confidence:** 3

**Review:**

Summary: This paper looks at designing backdoor attacks in federated learning system with an emphasis on persistence and stealth.The authors propose a backdoor attacking algorithm that selectively targets some layers in a neural network. This approach is based on the observation that some neural network layers converge quickly during training and subsequent training steps don't alter their weights much.  By embedding backdoor attacks after layers have converged, the attack remains persistent over time.

Strengths: This research is well motivated as federated learning systems are being deployed more widely. Devising novel attack schemes contribute to a deeper understanding of the vulnerabilities in these systems and informs potential defenses. The insight that updating inactive layers can yield more durable backdoors is both intuitive and interesting.

Questions:
1. I am not entirely clear on the gradient update step. For poisoned data, is projected gradient descent applied only to the inactive layers so that they alone are updated?
2. If inactive layers are typically unaffected by updates, how does the projection significantly alter them to embed the backdoor effectively?